# Inhibition of microRNA-33b in humanized mice ameliorates nonalcoholic steatohepatitis

Sawa Miyagawa[1], Takahiro Horie[1], Tomohiro Nishino[1], Satoshi Koyama[1], Toshimitsu Watanabe[1], Osamu Baba[1], Tomohiro Yamasaki[1], Naoya Sowa[2], Chiharu Otani[1], Kazuki Matsushita[1], Hidenori Kojima[1], Masahiro Kimura[1], Yasuhiro Nakashima[1], Satoshi Obika[3,4], Yuuya Kasahara[3,4], Jun Kotera[5], Kozo Oka[5], Ryo Fujita[5], Takashi Sasaki[5], Akihiro Takemiya[5], Koji Hasegawa[2], Takeshi Kimura[1], Koh Ono[1]

**Nonalcoholic steatohepatitis (NASH) can lead to cirrhosis and hepatocellular carcinoma in their advanced stages; however, there are currently no approved therapies. Here, we show that microRNA (miR)-33b in hepatocytes is critical for the development of NASH. miR-33b is located in the intron of sterol regulatory element–binding transcription factor 1 and is abundantly expressed in humans, but absent in rodents. miR-33b knock-in (KI) mice, which have a miR-33b sequence in the same intron of sterol regulatory element–binding transcription factor 1 as humans and express miR-33b similar to humans, exhibit NASH under high-fat diet feeding. This condition is ameliorated by hepatocyte-specific miR-33b deficiency but unaffected by macrophage-specific miR-33b deficiency. Anti-miR-33b oligonucleotide improves the phenotype of NASH in miR-33b KI mice fed a Gubra Amylin NASH diet, which induces miR-33b and worsens NASH more than a high-fat diet. Anti-miR-33b treatment reduces hepatic free cholesterol and triglyceride accumulation through up-regulation of the lipid metabolism–related target genes. Furthermore, it decreases the expression of fibrosis marker genes in cultured hepatic stellate cells. Thus, inhibition of miR-33b using nucleic acid medicine is a promising treatment for NASH.**

## Introduction

Over the past few decades, nonalcoholic fatty liver disease (NAFLD) has become the most common chronic liver disease (worldwide prevalence: ~25% of the adult population) and is now known to be associated with the components of metabolic syndrome (Younossi et al, 2016, 2018). Cirrhosis caused by a fatty liver was reported about 20 yr ago, but the term nonalcoholic steatohepatitis (NASH) was coined by Ludwig et al in 1980 (Thaler, 1962; Ludwig et al, 1980). In the United States, 21% of patients with NAFLD suffer from NASH; accordingly, the prevalence of NASH in the population of the United States is estimated at ~4%. Moreover, the prevalence of NAFLD/NASH is increasing, and these serious diseases can lead to advanced liver diseases, such as liver cirrhosis and hepatocellular carcinoma (Powell et al, 2021), but there are no approved therapies. NASH is also the second most common indication for liver transplantation in the United States after chronic hepatitis C (Haldar et al, 2019). Therefore, identifying the patients with NAFLD most at risk of developing NASH and finding a treatment for NAFLD/NASH are priorities for researchers and clinicians.

MicroRNA (miR)-33a has been found to play important roles in cholesterol metabolism by targeting the ATP-binding subfamily A member 1 (ABCA1) cholesterol transporter (Horie et al, 2010a; Najafi-Shoushtari et al, 2010; Rayner et al, 2010). miR-33a deficiency or inhibition in mice has been shown to increase cellular cholesterol export through up-regulation of ABCA1, thereby elevating blood high-density lipoprotein cholesterol (HDL-C) levels and inhibiting atherosclerosis (Rayner et al, 2011b; Horie et al, 2012; Rotllan et al, 2013). Inhibition of miR-33a in mice has also been shown to reduce plaque inflammation and prevent atherosclerosis, in part by promoting M2 macrophage polarization and Treg induction (Ouimet et al, 2015). On the contrary, the inhibition of liver miR-33a increased the blood HDL-C concentration and reversed the cholesterol transport capacity of mice fed a normal diet but did not reduce atherosclerotic plaque size under hyperlipidemic conditions where miR-33a expression was suppressed (Marquart et al, 2013). miR-33a has also been shown to repress target genes involved in other metabolic functions, including carnitine palmitoyltransferase 1A (CPT1A) and carnitine O-octanoyltransferase (CROT) for fatty acid metabolism, insulin receptor substrate 2 (IRS2) for insulin signaling, and protein kinase AMP-activated catalytic subunit alpha 1 (PRKAA1) and peroxisome proliferator–activated receptor gamma

[1]Department of Cardiovascular Medicine, Graduate School of Medicine, Kyoto University, Kyoto, Japan [2]Division of Translational Research, National Hospital Organization, Kyoto Medical Center, Kyoto, Japan [3]Graduate School of Pharmaceutical Sciences, Osaka University, Osaka, Japan [4]Center for Drug Design Research, National Institutes of Biomedical Innovation, Health and Nutrition, Osaka, Japan [5]Sohyaku. Innovative Research Division, Mitsubishi Tanabe Pharma Corporation, Shonan Health Innovation Park, Fujisawa-shi, Japan

Correspondence: kohono@kuhp.kyoto-u.ac.jp; thorie@kuhp.kyoto-u.ac.jp

coactivator 1 alpha (*PGC1α*) for mitochondrial function (Davalos et al, 2011; Karunakaran et al, 2015; Ouimet et al, 2015; Tang et al, 2017), as well as suppress sirtuin 6 (*SIRT6*), a NAD$^+$-dependent deacetylase, and induce inflammation in macrophages (He et al, 2017). The results of previous studies suggest that miR-33 may exacerbate glucose and lipid metabolism and inflammation in the liver under conditions of high-fat diet (HFD) loading. Indeed, experiments on liver-specific miR-33a–deficient mice revealed that their glucose homeostasis is improved and the development of fibrosis and inflammation is suppressed (Price et al, 2021). These experiments focused on the function of miR-33a, because mice have only miR-33a in the intron of sterol regulatory element–binding transcription factor (*Srebf*) 2. However, in humans, in addition to miR-33a, miR-33b is present in the intron of *SREBF1* and is highly expressed in the liver. Because *SREBF1* is known to be induced in the NAFLD/NASH liver and be involved in its formation (Chen et al, 2004; Ferre & Foufelle, 2010), its intronic miR-33b is also expected to be induced and exert some function in this condition, but it remains unclear.

To clarify the role of miR-33b in vivo, we conducted experiments using mice with miR-33b knock-in (KI) in the intron of *Srebf1* (miR-33b KI mice), similar to the condition in humans (Horie et al, 2014). These humanized miR-33b KI mice express miR-33b along with its host gene *Srebf1* at the physiological level. With their WT littermates, miR-33b KI mice were fed a 45 kcal% HFD, after which the miR-33b KI mice exhibited NAFLD/NASH. Thus, these mice were considered a NASH model, which was improved by hepatocyte-specific miR-33b deficiency but unaffected by macrophage-specific miR-33b deficiency. Under Gubra Amylin NASH (GAN) diet–loading conditions, which are considered to increase the severity of liver damage, we examined the effects of anti-miRNA oligonucleotides (AMOs) that specifically inhibit miR-33a and miR-33b (Yamasaki et al, 2022). We found that anti-miR-33b was potent in terms of reducing the accumulation of free cholesterol (F-Cho) and triglycerides in the liver and improving fibrosis.

# Results

## miR-33b KI mice exhibit the NASH phenotype under HFD feeding conditions

Male miR-33b KI mice (miR-33b$^{+/+}$) and WT littermate mice (miR-33b$^{-/-}$) were fed a 45 kcal% HFD for 12 wk starting at 8 wk of age (Fig 1A). There was no change in the body weight or food intake of the mice (Fig S1A and B), nor any change in their liver/body weight ratio (Fig S1C). Blood sample analysis revealed a significant reduction in total cholesterol (T-Cho) and HDL-C levels in the blood of miR-33b KI mice, as reported previously (Horie et al, 2014). Low-density lipoprotein cholesterol (LDL-C) levels were also reduced in these blood samples. In contrast, the levels of liver enzymes such as aspartate transaminase (AST), alanine transaminase (ALT), alkaline phosphatase (ALP), and total bilirubin (T-BIL) were significantly higher in miR-33b KI mice than in WT mice, suggesting the deterioration of liver function in miR-33b KI mice under HFD feeding (Table 1). Consistent with the serum data, HE staining revealed fat

accumulation, inflammatory cell infiltration, and hepatocyte damage (balloon-like hepatocyte enlargement) in the livers of miR-33b KI mice (Fig 1B). The livers of miR-33b KI mice also exhibited significantly elevated T-Cho content and a nonsignificant increase in triglyceride levels (Fig 1C). Analysis of the target genes of miR-33 showed that the mRNA expression of *Cpt1a* was significantly decreased and the expression of other miR-33 target genes tended to be decreased in the livers of miR-33b KI mice (Fig 1D). In addition, the protein levels of ABCA1 and CPT1A were significantly reduced in the livers of miR-33b KI mice (Fig 1E and F). However, the protein level of CROT did not differ between miR-33b KI and WT mice (Fig 1E and F). Conversely, the expression levels of inflammatory genes and fibrosis-related genes, namely, *Tnf* and *Col1a1*, were significantly elevated in the livers of miR-33b KI mice (Fig 1G), and the protein level of COL1A1 was also elevated (Fig 1H and I). Masson's trichrome staining revealed the presence of severe fibrosis in the livers of miR-33b KI mice, that is, the histological signature of NASH (Fig 1J). Furthermore, hepatic metabolomics analysis of miR-33a knockout and miR-33b KI mice, as well as their littermate control mice, showed that clusters of lipid components were altered in the livers of miR-33b KI mice (Fig S1D). Thus, when subjected to a HFD burden, miR-33b KI mice exhibited the NASH phenotype accompanied by lipid regulatory gene expression changes and lipid accumulation in the liver.

## miR-33b in hepatocytes but not macrophages influences pathological deterioration

To determine the cells and organs responsible for the NASH phenotype owing to their miR-33b expression, we generated mice in which loxP-flanked miR-33b was inserted into intron 16 of *Srebf1* (miR-33b$^{fl/fl}$ KI mice) (Fig S2A). Southern blotting of ES cells and PCR analysis of the tail genome indicated that homologous recombination was successful (Fig S2B and C). Moreover, this strategy did not alter the splicing of intron 16 of *Srebf1*, as confirmed by sequencing (Fig S2D). To confirm the Cre-mediated deletion of miR-33b, miR-33b$^{fl/fl}$ KI mice were crossed with *Ayu1*-Cre mice, which express Cre recombinase in multiple tissues including the germ line (Morita et al, 2003). Although these mice had similar body weights, miR-33b levels were significantly reduced in several organs, including the liver, epididymal white adipose tissue, heart, and skeletal muscle, without any change in miR-33a expression levels (Fig S3A–C). Hepatic *Abca1* expression and serum HDL-C levels were higher in these mice than in miR-33b$^{fl/fl}$ KI mice (Fig S3D and Table S1). To evaluate the contribution of miR-33b in hepatocytes, we crossed miR-33b$^{fl/fl}$ KI mice with *Alb*-Cre mice. miR-33b expression levels were significantly lower in the livers of *Alb*-Cre/miR-33b$^{fl/fl}$ KI mice than in those of miR-33b$^{fl/fl}$ KI mice, although miR-33a expression levels did not differ between these mice (Fig S4A). However, serum T-Cho, HDL-C, and LDL-C levels were all elevated in *Alb*-Cre/miR-33b$^{fl/fl}$ KI mice (Table S2). When these mice were fed a 45% HFD from the age of 8 wk for 12 wk, no change in body weight gain, food intake, or liver/body weight ratio was found (Fig S4B–E). However, compared with miR-33b$^{fl/fl}$ KI mice, *Alb*-Cre/miR-33b$^{fl/fl}$ KI mice exhibited a marked decrease in steatosis and balloon-like cells in the liver according to HE staining (Fig 2A). Serum analysis also showed that AST, ALT, and lactate dehydrogenase (LDH) levels

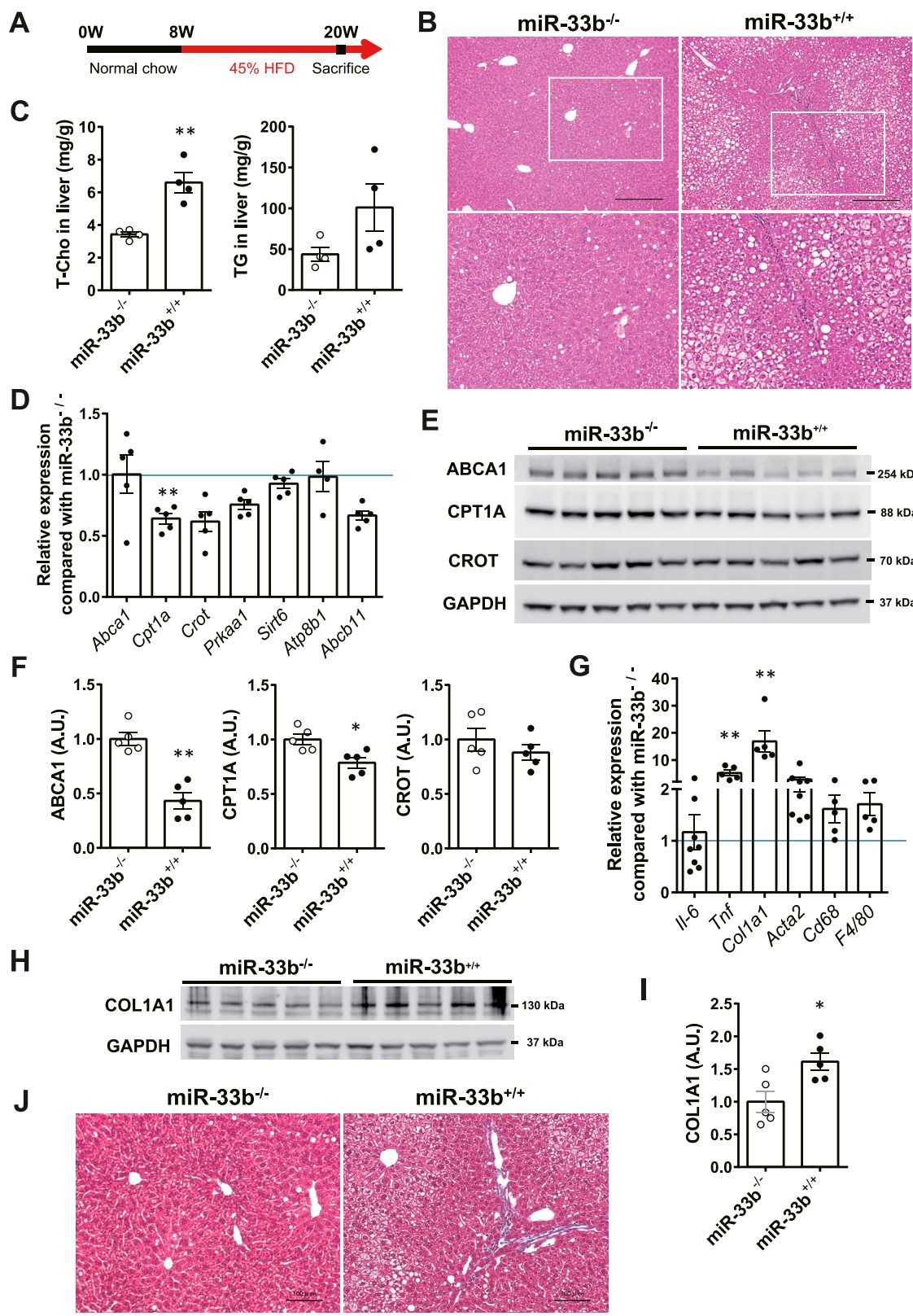

**Figure 1. miR-33b KI mice exhibit the NASH phenotype under HFD feeding.**

**(A)** Scheme of the HFD feeding protocol. **(B)** Representative microscopic images of HE staining of the liver of miR-33b⁻/⁻ and miR-33b⁺/⁺ mice fed the HFD. Scale bars: 300 μm (upper). **(C)** Total cholesterol and triglyceride levels in the livers of miR-33b⁻/⁻ and miR-33b⁺/⁺ mice fed the HFD. n = 4 mice per group; **P < 0.01, unpaired t test. **(D)** Relative expression levels of miR-33 target genes in the livers of miR-33b⁻/⁻ and miR-33b⁺/⁺ mice fed the HFD. n = 4–5 mice per group; **P < 0.01, unpaired t test.

**Table 1.  Serum data of miR-33b$^{-/-}$ and miR-33b$^{+/+}$ mice fed a 45% HFD.**

| | miR-33b$^{-/-}$ | miR-33b$^{+/+}$ | |
|---|---|---|---|
| TP (g/dl) | 5.00 ± 0.08 | 4.81 ± 0.13 | |
| ALB (g/dl) | 3.16 ± 0.05 | 3.13 ± 0.08 | |
| AST (IU/l) | 46.3 ± 2.9 | 91.3 ± 15.6 | ** |
| ALT (IU/l) | 26.2 ± 4.3 | 60.6 ± 14.1 | * |
| ALP (IU/l) | 157.6 ± 8.3 | 213.1 ± 14.5 | ** |
| LDH (IU/l) | 492.7 ± 81.4 | 563.1 ± 101.6 | |
| T-BIL (mg/dl) | 0.074 ± 0.007 | 0.175 ± 0.043 | * |
| T-Cho (mg/dl) | 164.5 ± 6.6 | 84.40 ± 11.3 | *** |
| LDL-C (mg/dl) | 12.3 ± 1.3 | 6.7 ± 0.6 | ** |
| HDL-C (mg/dl) | 77.6 ± 1.0 | 46.4 ± 5.6 | *** |
| TG (mg/dl) | 18.3 ± 1.8 | 10.9 ± 1.0 | ** |
| NEFA (μEq/l) | 612.1 ± 34.2 | 602.7 ± 44.1 | |
| CRE (mg/dl) | 0.09 ± 0.007 | 0.08 ± 0.004 | |
| BUN (mg/dl) | 17.6 ± 0.57 | 16.9 ± 0.58 | |

Male mice were fed a 45% HFD from the age of 8 wk for 12 wk. Values are the mean ± S.E.M., n = 10, 11 each; *$P < 0.05$, **$P < 0.01$, ***$P < 0.001$, unpaired $t$ test.

were all significantly reduced in *Alb*-Cre/miR-33b$^{fl/fl}$ KI mice, indicating the amelioration of liver dysfunction (Table 2). Serum T-Cho and HDL-C levels were significantly elevated in these mice (Table 2). ABCA1 levels were significantly higher (Fig 2B and C), and *Col1a1* and *Acta2* levels were significantly lower and *Tnf* levels tended to be reduced in the livers of *Alb*-Cre/miR-33b$^{fl/fl}$ KI mice than in those of miR-33b$^{fl/fl}$ KI mice (Fig 2D). Picro-Sirius red staining and its quantification indicated that the fibrosis area in the livers of *Alb*-Cre/miR-33b$^{fl/fl}$ KI mice was significantly decreased (Fig 2B and E). Moreover, hepatic T-Cho levels were significantly lower and triglyceride levels tended to be lower in *Alb*-Cre/miR-33b$^{fl/fl}$ KI mice than in miR-33b$^{fl/fl}$ KI mice (Fig 2F and G). To evaluate the contribution of miR-33b in macrophages, we crossed miR-33b$^{fl/fl}$ KI mice with *LysM*-Cre mice. Successful recombination was confirmed via the analysis of peritoneal macrophages. miR-33b levels were significantly lower in the macrophages of *LysM*-Cre/miR-33b$^{fl/fl}$ KI mice than in those of miR-33b$^{fl/fl}$ KI mice, whereas miR-33a levels did not differ between these mice (Fig S5A). Similarly, serum data did not differ between the mice, other than a significant decrease in ALT levels in *LysM*-Cre/miR-33b$^{fl/fl}$ KI mice at 8 wk of age (Table S3). The mice were fed a 45% HFD from the age of 8 wk for 12 wk, and no change in body weight gain and body weight at euthanasia (Fig S5B and C) or in serum lipid and liver enzyme levels (Table S4) was found between miR-33b$^{fl/fl}$ KI mice and *LysM*-Cre/miR-33b$^{fl/fl}$ KI mice. The expression levels of lipid regulatory genes, inflammatory genes, and fibrosis-related genes were similar between miR-33b$^{fl/fl}$

KI mice and *LysM*-Cre/miR-33b$^{fl/fl}$ KI mice (Fig S5D), and HE staining revealed no difference in their liver histology (Fig S5E). These data suggest that miR-33b in hepatocytes but not macrophages is responsible for the HFD-induced NASH phenotype in miR-33b KI mice.

### Anti-miR-33b treatment ameliorates GAN diet–induced liver dysfunction

To investigate the possibility that the inhibition of miR-33 could ameliorate the NASH phenotype, we fed male miR-33b KI mice the GAN diet rather than a HFD to induce NASH. The GAN diet is high in cholesterol (2%) and fat (40 kcal%) and contains fructose and palm oil; it is used to induce NASH because it causes a higher degree of liver inflammation and fibrosis than the normal HFD (Boland et al, 2019). We previously developed AMOs capable of inhibiting miR-33a and miR-33b individually in vivo (Yamasaki et al, 2022), and these were applied in the present study. Nucleic acids were administered to mice from 8 wk of age when their NC diet either continued or was changed from NC to GAN (Fig 3A). NEG, anti-miR-33a, anti-miR-33b, and anti-miR-33a + b oligonucleotides were subcutaneously injected (10 mg/kg nucleic acid doses [5 mg/kg of each oligonucleotide in the anti-miR-33a + b treatment]) once every 2 wk, with seven injections administered in total. 1 wk after the last dose with mice at 21 wk of age, the mice were euthanized and their blood and organs were collected. Body weight gain and body weight at euthanasia were similar among the treatment groups, although the anti-miR-33b group tended to be slightly underweight (Fig S6A and B). Liver/body weight ratio was significantly increased by the GAN diet but did not differ among the AMO treatment groups under the GAN diet condition (Fig S6C). Epididymal white adipose tissue tended to be increased by the GAN diet but decreased by about 0.8-fold in the anti-miR-33b group (Fig S6D). To assess the contribution of miR-33a and miR-33b, the copy numbers of each miRNA in the liver were calculated using standard oligonucleotides. The expression of miR-33a in the liver was significantly increased by GAN diet feeding but suppressed in the anti-miR-33a, anti-miR-33b, and anti-miR-33a + b groups (Fig 3B, left). The expression of miR-33b was also significantly increased by the GAN diet but suppressed in the anti-miR-33b and anti-miR-33a + b groups (Fig 3B, middle). The copy number of miR-33b was sixfold higher than that of miR-33a in the NEG group, indicating the importance of miR-33b in the liver. Thus, the total copy number of miR-33 was significantly increased by the GAN diet, whereas this effect was significantly reversed in the anti-miR-33b and anti-miR-33a + b groups (Fig 3B, right). The expression of *Srebf1*, the host gene of miR-33b, was significantly increased by the GAN diet but decreased in the anti-miR-33b group, whereas the expression of *Srebf2*, the host gene of miR-33a, did not differ significantly among the treatment groups (Fig 3C). HE staining revealed marked fat accumulation, inflammatory cell infiltration,

---

**(E)** Western blotting analysis of ABCA1, CPT1A, and CROT expression in the livers of miR-33b$^{-/-}$ and miR-33b$^{+/+}$ mice fed the HFD. GAPDH was used as a loading control. n = 5 mice per group. **(F)** Densitometry of hepatic ABCA1, CPT1A, and CROT. n = 5 mice per group; *$P < 0.05$ and **$P < 0.01$, unpaired $t$ test. **(G)** Relative expression levels of inflammatory genes and fibrosis-related genes in the livers of miR-33b$^{-/-}$ and miR-33b$^{+/+}$ mice fed the HFD. n = 5–8 mice per group; **$P < 0.01$, unpaired $t$ test. **(H)** Western blotting analysis of COL1A1 expression in the livers of miR-33b$^{-/-}$ and miR-33b$^{+/+}$ mice fed the HFD. GAPDH was used as a loading control. n = 5 mice per group. **(I)** Densitometry of hepatic COL1A1. n = 5 mice per group; *$P < 0.05$, unpaired $t$ test. **(J)** Representative microscopic images of Masson's trichrome staining of the livers of miR-33b$^{-/-}$ and miR-33b$^{+/+}$ mice fed the HFD. Scale bars: 100 μm.

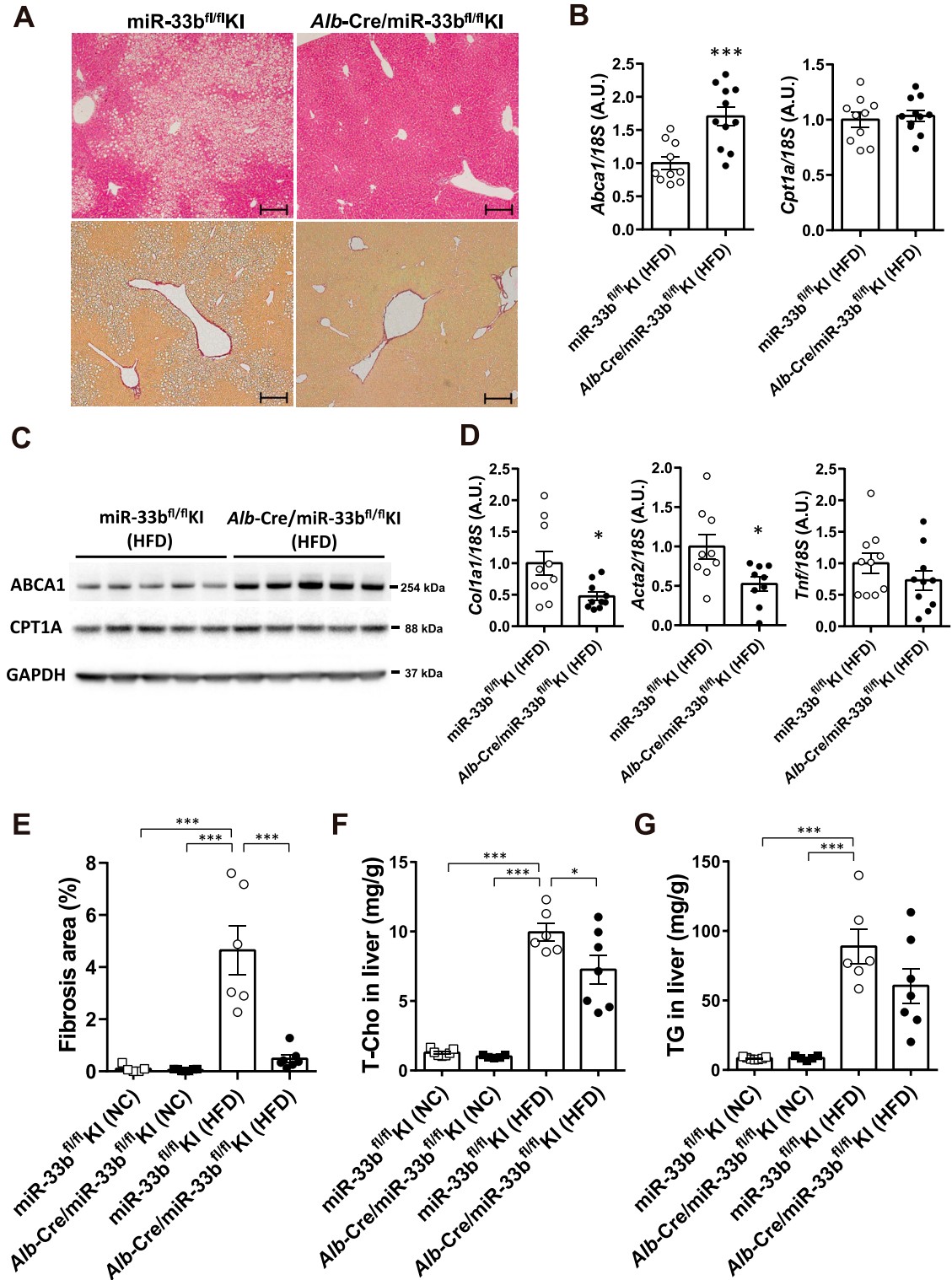

**Figure 2. HFD-induced NASH phenotype is ameliorated in Alb-Cre/miR-33b^fl/fl KI mice.**
**(A)** Representative microscopic images of HE (upper) and Picro-Sirius red (lower) staining of the livers of miR-33b^fl/fl KI and *Alb*-Cre/miR-33b^fl/fl KI mice fed the HFD. Scale bars: 200 μm. **(B)** Relative expression levels of *Abca1* and *Cpt1a* in the livers of miR-33b^fl/fl KI and *Alb*-Cre/miR-33b^fl/fl KI mice fed the HFD. n = 10–11 mice per group; ***P < 0.001, unpaired t test. **(C)** Western blotting analysis of ABCA1 and CPT1A expression in the livers of miR-33b^fl/fl KI and *Alb*-Cre/miR-33b^fl/fl KI mice fed the HFD. GAPDH was used as a loading control. n = 5 mice per group. **(D)** Relative expression levels of *Col1a1*, *Acta2*, and *Tnf* in the livers of miR-33b^fl/fl KI and *Alb*-Cre/miR-33b^fl/fl KI mice fed the HFD. n = 9–10 mice per group; *P < 0.05, unpaired t test. **(E)** Quantification of the fibrosis area in the liver sections of miR-33b^fl/fl KI and *Alb*-Cre/miR-33b^fl/fl KI mice fed NC or the HFD according to Picro-Sirius red staining. n = 6–7 mice per group; ***P < 0.001, one-way ANOVA with Tukey's post hoc test. **(F)** Total cholesterol content in the

and hepatocyte damage (balloon-like hepatocyte enlargement) because of GAN diet feeding, whereas these effects were ameliorated in the anti-miR-33b and anti-miR-33a + b groups (Fig 3D). In addition, serum analysis revealed marked increases in the levels of liver enzymes such as AST, ALT, ALP, LDH, ChE, T-BIL, and total bile acid because of GAN diet feeding, whereas these effects were significantly ameliorated by the anti-miR-33b and anti-miR-33a + b treatments and partly ameliorated by the anti-miR-33a treatment (Table 3). Overall, these data indicate that anti-miR-33 treatment, especially anti-miR-33b treatment, ameliorates GAN diet–induced liver dysfunction in miR-33b KI mice.

### Anti-miR-33b treatment improves the lipid profile in the serum and liver under GAN diet feeding

The expression levels of hepatic *Abca1* and *Cpt1a*, the target genes of miR-33, were significantly higher in the anti-miR-33b group than in the NEG group under GAN diet feeding (Fig 4A), and the anti-miR-33a + b group showed the same tendency as the anti-miR-33b group. The up-regulation of ABCA1 and CPT1A protein levels was also confirmed in the anti-miR-33b group (Fig 4B and C), although CROT expression was not changed at the mRNA or protein level. In serum lipids, T-Cho and HDL-C levels were increased by GAN diet feeding, and these levels were further elevated in the anti-miR-33b and anti-miR-33a + b groups relative to the NEG group (Table 3). Therefore, the increase in cholesterol was considered to be mainly due to the elevation of HDL-C content. Moreover, F-Cho levels were also increased by the GAN diet, and these levels were further increased in the anti-miR-33b and anti-miR-33a + b groups relative to the NEG group (Table 3). In contrast, triglyceride levels were significantly lower in the anti-miR-33a + b group than in the NEG group (Table 3). The lipids in the liver were quantified using the Folch method (Folch et al, 1957), and T-Cho content was significantly increased by ~9.5-fold by the GAN diet; however, T-Cho levels tended to be lower in anti-miR-33b–treated mice than in NEG-treated mice under GAN diet feeding. F-Cho content was also significantly increased by ~2.4-fold by the GAN diet, and F-Cho levels were significantly lower in the anti-miR-33b group than in the NEG group under GAN diet feeding. The triglyceride level was significantly increased by 3.1-fold by the GAN diet, and triglyceride content tended to be lower in the anti-miR-33b group than in the NEG group (Fig 4D). We also measured cholesterol crystals in the liver using a polarizing filter (Fig 4E), finding that cholesterol crystal content was significantly increased by the GAN diet but significantly decreased by the anti-miR-33a, anti-miR-33b, and anti-miR-33a + b treatments relative to the NEG treatment under GAN diet feeding (Fig 4F). ABCA1 is known to transport cholesterol, especially F-Cho, into the blood vessels; thus, the decrease in liver F-Cho content and the increase in serum HDL-C or F-Cho content may be due to the up-regulation of hepatic ABCA1. Anti-miR-33 treatment, especially anti-miR-33b treatment, improved the lipid profile in the serum and liver, which explains the amelioration of the GAN diet–induced

**Table 2.  Serum data of miR-33$^{fl/fl}$ KI and *Alb*-Cre/miR-33b$^{fl/fl}$ KI mice fed a 45% HFD.**

|  | miR-33b$^{fl/fl}$ KI | *Alb*-Cre/miR-33b$^{fl/fl}$ KI |  |
|---|---|---|---|
| TP (g/dl) | 4.33 ± 0.20 | 4.43 ± 0.08 |  |
| ALB (g/dl) | 2.73 ± 0.08 | 2.77 ± 0.04 |  |
| AST (IU/l) | 105.8 ± 27.4 | 45.7 ± 2.6 | * |
| ALT (IU/l) | 63.3 ± 16.2 | 26.7 ± 4.9 | * |
| ALP (IU/l) | 237.5 ± 51.3 | 213.1 ± 14.5 |  |
| LDH (IU/l) | 663.5 ± 100.1 | 420.4 ± 55.6 | * |
| T-BIL (mg/dl) | 0.241 ± 0.103 | 0.075 ± 0.007 |  |
| TBA (μmol/l) | 29.8 ± 20.0 | 1.43 ± 0.20 |  |
| T-Cho (mg/dl) | 71.8 ± 12.8 | 139.1 ± 9.2 | ** |
| LDL-C (mg/dl) | 5.5 ± 0.7 | 3.2 ± 1.3 | * |
| HDL-C (mg/dl) | 38.8 ± 7.9 | 64.6 ± 8.0 | ** |
| TG (mg/dl) | 25.1 ± 4.0 | 23.9 ± 4.2 |  |
| NEFA (μEq/l) | 428.0 ± 41.2 | 433.6 ± 30.8 |  |

Male mice were fed a 45% HFD from the age of 8 wk for 12 wk. Values are the mean ± S.E.M., n = 6, 7 each; *$P$ < 0.05 and **$P$ < 0.01, unpaired $t$ test.

NASH phenotype in miR-33b KI mice via the up-regulation of ABCA1 and other lipid regulatory genes.

### Anti-miR-33b treatment ameliorates GAN diet–induced inflammation and fibrosis

We evaluated inflammation and fibrosis in the liver, finding that the expression of *Tnf* was significantly increased by the GAN diet but suppressed by the anti-miR-33a, anti-miR-33b, and anti-miR-33a + b treatments (Fig 5A, left). In addition, F4/80 staining and MPO staining were also performed to determine the effect of anti-miR-33a and anti-miR-33b on inflammation. The results confirmed that the GAN diet increased F4/80- and MPO-positive cells in the liver, whereas administration of anti-miR-33a and anti-miR-33b decreased them (Fig S7A–D). Likewise, the expression of *Col1a1* was significantly increased by the GAN diet but suppressed by the anti-miR-33a, anti-miR-33b, and anti-miR-33a + b treatments (Fig 5A, middle). A similar expression pattern was observed for *Acta2* expression (Fig 5A, right). The presence of αSMA, the protein encoded by *Acta2*, was confirmed via immunofluorescence staining (Fig 5B). The αSMA-positive area was significantly increased by the GAN diet but significantly suppressed by the anti-miR-33a, anti-miR-33b, and anti-miR-33a + b treatments (Fig 5C). Fibrosis of the liver was also evaluated using Picro-Sirius red staining (Fig 5D), and the fibrosis area was significantly reduced in the anti-miR-33b treatment group (Fig 5E). Histological changes were further assessed using the NAFLD activity score (NAS) and NASH fibrosis stage score (Brunt et al, 1999; Kleiner et al, 2005). The NAS scores the degree of fatness (0–3 points), parenchymal inflammatory stage (0–3 points), and

---

livers of miR-33b$^{fl/fl}$ KI and *Alb*-Cre/miR-33b$^{fl/fl}$ KI mice fed NC or the HFD. n = 6–7 mice per group; *$P$ < 0.05 and ***$P$ < 0.001, one-way ANOVA with Tukey's post hoc test.
**(G)** Triglyceride content in the livers of miR-33b$^{fl/fl}$ KI and *Alb*-Cre/miR-33b$^{fl/fl}$ KI mice fed NC or the HFD. n = 6–7 mice per group; ***$P$ < 0.001, one-way ANOVA with Tukey's post hoc test.

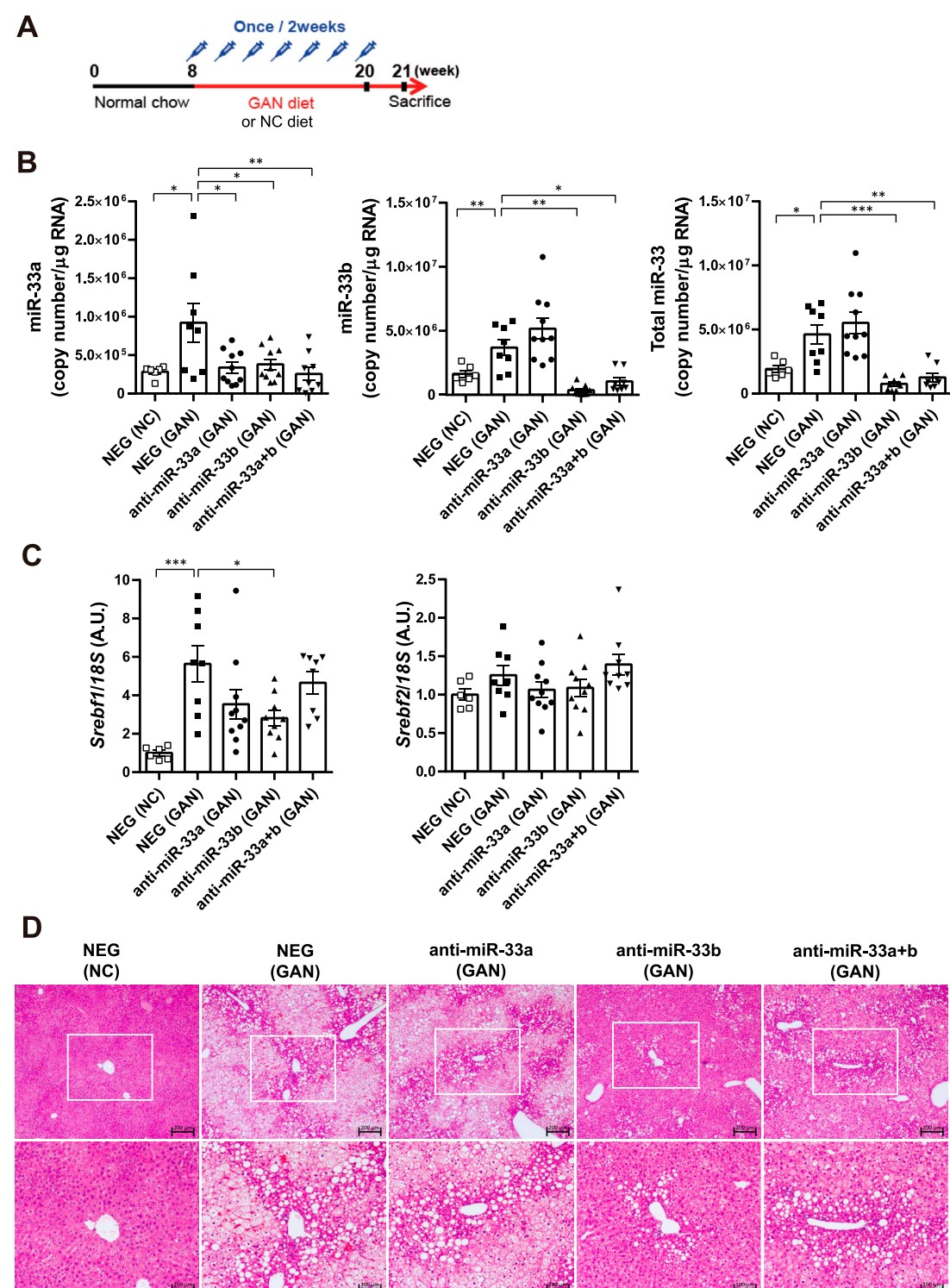

**Figure 3. Anti-miR-33 treatment ameliorates the GAN diet–induced NASH phenotype.**
**(A)** Scheme of the anti-miRNA oligonucleotide administration protocol. **(B)** Absolute copy numbers of miR-33a, miR-33b, and total miR-33 in the livers of the indicated groups. n = 6–10 samples per group; *P < 0.05, **P < 0.01, and ***P < 0.001, one-way ANOVA with Tukey's post hoc test. **(C)** Relative expression levels of *Srebf1* and *Srebf2* in the livers of the indicated groups. n = 6–10 samples per group; *P < 0.05 and ***P < 0.001, one-way ANOVA with Tukey's post hoc test. **(D)** Representative microscopic images of HE staining of the liver of the indicated groups. Scale bars: 200 μm (upper) and 100 μm (lower).

**Table 3. Serum data of miR-33b$^{+/+}$ mice treated with AMOs.**

| Diet | NC | GAN | | GAN | | GAN | | GAN | |
|---|---|---|---|---|---|---|---|---|---|
| AMOs | NEG | NEG | | anti-miR-33a | | anti-miR-33b | | anti-miR-33a + b | |
| TP (g/dl) | 5.06 ± 0.10 | 5.24 ± 0.09 | | 5.31 ± 0.15 | | 5.30 ± 0.15 | | 5.13 ± 0.06 | |
| ALB (g/dl) | 3.33 ± 0.10 | 3.23 ± 0.08 | | 3.21 ± 0.11 | | 3.23 ± 0.06 | | 3.02 ± 0.04 | |
| AST (IU/l) | 102.0 ± 13.4 | 725.6 ± 157.3 | *** | 423.4 ± 42.1 | | 243.8 ± 72.0 | ## | 260.4 ± 42.9 | ## |
| ALT (IU/l) | 67.7 ± 24.8 | 749.1 ± 215.0 | ** | 349.0 ± 53.1 | | 234.1 ± 68.9 | # | 240.6 ± 46.8 | # |
| ALP (IU/l) | 314.5 ± 21.3 | 596.4 ± 63.19 | ** | 657.0 ± 57.7 | | 357.3 ± 41.9 | ## | 370.1 ± 25.4 | # |
| LDH (IU/l) | 1,505 ± 49.8 | 4,181 ± 654.0 | *** | 2,777 ± 213.7 | # | 2007 ± 204.1 | ### | 1,493 ± 178.4 | ### |
| ChE (IU/l) | 34.7 ± 1.1 | 49.9 ± 2.7 | *** | 46.4 ± 1.2 | | 35.7 ± 2.0 | ### | 38.3 ± 1.6 | ### |
| T-BIL (mg/dl) | 0.100 ± 0.006 | 0.402 ± 0.113 | * | 0.336 ± 0.086 | | 0.076 ± 0.008 | ## | 0.070 ± 0.008 | ## |
| TBA (μmol/l) | 15.0 ± 9.17 | 83.5 ± 29.6 | * | 51.9 ± 14.0 | | 6.0 ± 0.9 | ## | 4.6 ± 0.8 | ## |
| Fe (μg/dl) | 114.7 ± 9.3 | 192.4 ± 8.2 | *** | 195.1 ± 9.6 | | 202.6 ± 11.8 | | 177.4 ± 9.7 | |
| T-Cho (mg/dl) | 63.8 ± 3.6 | 114.5 ± 9.8 | ** | 121.9 ± 12.6 | | 186.2 ± 8.6 | ### | 166.5 ± 4.5 | ## |
| LDL-C (mg/dl) | 6.2 ± 0.3 | 17.9 ± 0.83 | *** | 17.1 ± 0.82 | | 23.0 ± 1.5 | # | 18.2 ± 1.1 | |
| HDL-C (mg/dl) | 44.3 ± 2.3 | 61.5 ± 7.5 | | 65.1 ± 8.5 | | 99.2 ± 3.5 | ### | 92.0 ± 2.0 | ## |
| F-CHO (mg/dl) | 7.8 ± 0.8 | 20.0 ± 0.9 | * | 26.4 ± 3.4 | | 38.4 ± 3.3 | ### | 30.5 ± 1.6 | # |
| TG (mg/dl) | 24.2 ± 2.5 | 18.9 ± 1.2 | | 18.1 ± 2.6 | | 15.7 ± 1.9 | | 9.5 ± 0.6 | # |
| NEFA (μEq/l) | 999.5 ± 65.5 | 1,036 ± 88.2 | | 1,192 ± 69.6 | | 1,065 ± 82.9 | | 962.5 ± 41.0 | |
| CRE (mg/dl) | 0.11 ± 0.010 | 0.12 ± 0.006 | | 0.13 ± 0.007 | | 0.13 ± 0.011 | | 0.12 ± 0.007 | |
| BUN (mg/dl) | 21.3 ± 1.09 | 23.4 ± 0.77 | | 23.53 ± 1.12 | | 23.7 ± 1.02 | | 22.9 ± 0.54 | |

Male miR-33b+/+ mice were fed NC or the GAN diet and injected with the indicated AMOs from the age of 8 wk for 12 wk. Values are the mean ± S.E.M., n = 6 for NC and n = 8–10 for the GAN diet each, one-way ANOVA with Turkey's post hoc test. *$P < 0.05$, **$P < 0.01$, and ***$P < 0.001$, compared with NEG (NC diet). #$P < 0.05$, ##$P < 0.01$, and ###$P < 0.001$, compared with NEG (GAN diet).

balloon-like hepatocyte enlargement (0–2 points). Compared with the NAS in the NEG group (mean: 6.9 points), the NAS was significantly improved in the anti-miR-33b group (mean: 3.0 points) and anti-33a + b group (mean: 4.3 points) (Fig 5F). The NASH fibrosis stage is scored on a scale of 0–4 according to the degree of fibrosis; compared with the NEG group (mean: 2.1), the NASH fibrosis stage was significantly improved in the anti-miR-33a group (mean: 1.3), anti-miR-33b group (mean: 0.22), and anti-33a + b group (mean: 0.75) (Fig 5G). Thus, the suppressive effects of inflammatory and fibrosis-related genes were most prominent after anti-miR-33b treatment.

### Anti-fibrotic effects of anti-miR-33b treatment in cultured hepatic stellate cells (HSCs)

Changes in the expression levels of fibrosis-related genes such as *Col1a1* and *α*SMA (*Acta2*) suggested that anti-miR-33 treatment affects HSCs. Therefore, we conducted in vitro experiments to determine whether anti-miR-33 treatment directly affects HSCs. Lipofection of anti-miR-33b significantly decreased miR-33b expression in the human HSC cell line LI90 (Murakami et al, 1995) (Fig 6A). In addition, miR-33b expression was induced by TGF-*β* stimulation in LI90 cells (Fig 6B). Anti-miR-33b treatment significantly increased the protein levels of ABCA1 and CPT1A, although it had no significant effect on *ABCA1*, *CPT1A*, and *CROT* mRNA expression (Fig 6C–E). Concurrently, significant reductions in *COL1A1*, *ACTA2*, and

*POSTN* expression levels were observed in anti-miR-33b–treated cells (Fig 6F).

## Discussion

In the present study, our four major findings were as follows: (i) miR-33b KI mice, in which miR-33b was knocked into the intron of *Srebf1* to mimic *SREBF1* in humans, showed NAFLD/NASH when fed a 45 kcal% HFD; (ii) based on analysis of cell-specific Cre/miR-33b$^{fl/fl}$ KI mice, miR-33b in hepatocytes but not macrophages was important for increasing the severity of NAFLD/NASH; (iii) under GAN diet–loaded conditions, anti-miR-33b AMO was effective for treating NAFLD/NASH as it suppressed liver F-Cho and triglyceride accumulation and improved fibrosis in miR-33b KI mice; and (iv) HSC miR-33b may also contribute to the exacerbation of NAFLD/NASH.

NAFLD is a chronic liver disease characterized by excessive lipid deposition ("hepatic lipidosis") within lipid droplets in hepatocytes in the absence of significant alcohol consumption, viral hepatitis, and other liver diseases (Chalasani et al, 2012). NAFLD ranges in histological severity from asymptomatic hepatic lipidosis to cirrhosis. Most patients with NAFLD have simple lipidosis without parenchymal inflammation or fibrosis, and such patients generally follow a benign clinical course from a hepatic standpoint and are

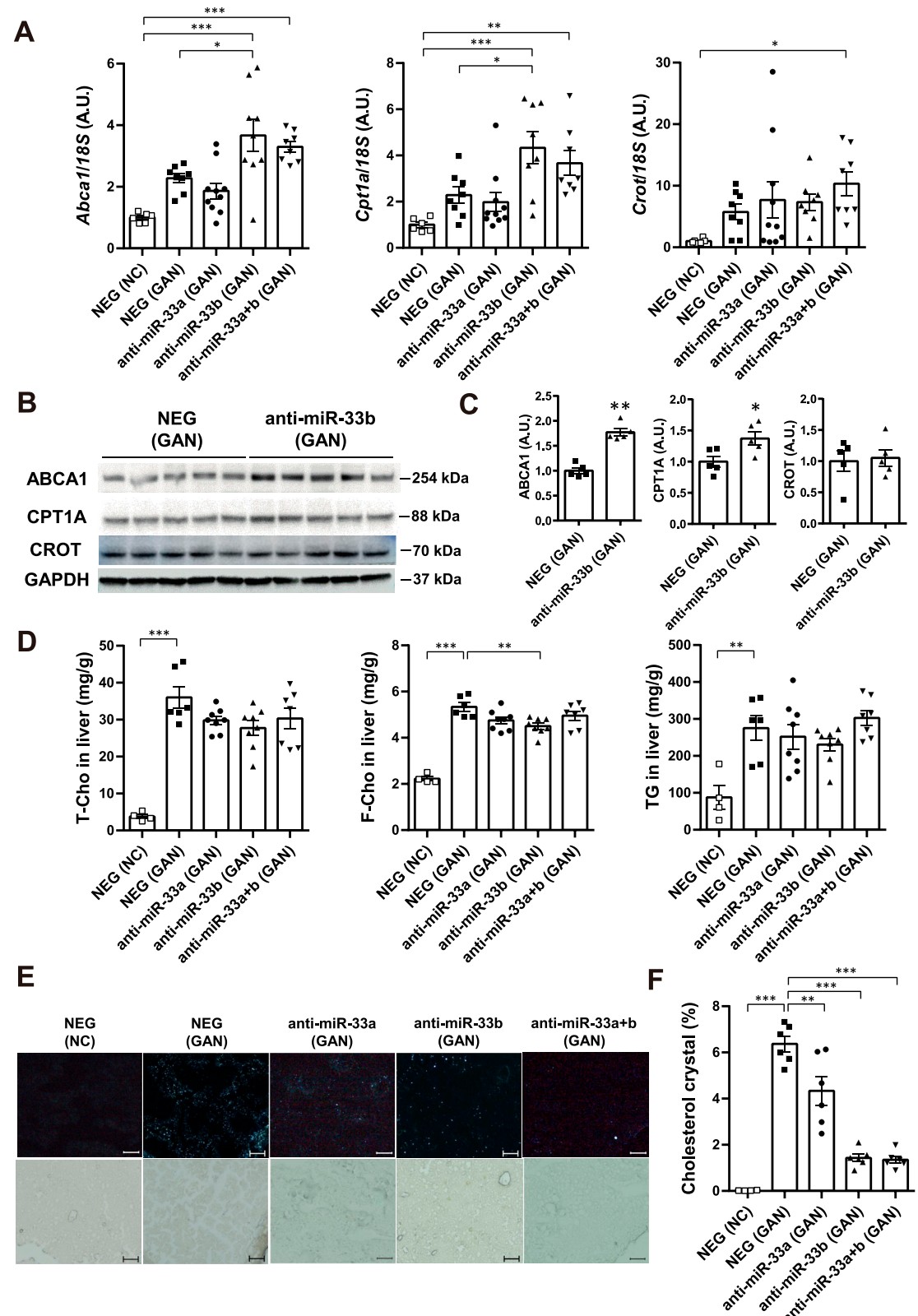

**Figure 4. Anti-miR-33 treatment with the GAN diet improves the lipid profile in the serum and liver.**
**(A)** Relative expression levels of *Abca1*, *Cpt1a*, and *Crot* in the livers of the indicated groups. n = 6–10 samples per group; *$P < 0.05$ and ***$P < 0.001$, one-way ANOVA with Tukey's post hoc test. **(B)** Western blotting analysis of ABCA1, CPT1A, and CROT expression in the livers of the indicated groups. GAPDH was used as a loading control. n = 5 samples per group. **(C)** Densitometry of hepatic ABCA1, CPT1A, and CROT. n = 5 mice per group; *$P < 0.05$ and **$P < 0.01$, unpaired *t* test. **(D)** Total cholesterol, free cholesterol,

unlikely to develop cirrhosis (Matteoni et al, 1999). In contrast, approximately 10–30% of patients with NAFLD develop NASH (Williams et al, 2011), which is an important disease because it can progress to cirrhosis, liver failure, and hepatocellular carcinoma (Matteoni et al, 1999; Bugianesi et al, 2002).

The results of several studies suggest that lipotoxicity because of hepatic F-Cho overload is the mechanistic driver of the necroinflammation and fibrosis that characterize NASH in many animal models and in some patients with NASH (Bugianesi et al, 2002; Yamaguchi et al, 2007; Wouters et al, 2010; Gan et al, 2014; Ioannou, 2016; Song et al, 2021). Hyperinsulinemia resulting from diet, lifestyle, obesity, major genetic polymorphisms, and insulin resistance is a pivotal factor leading to abnormal cholesterol signaling and F-Cho accumulation in hepatocytes (Xie et al, 2009; Van Rooyen et al, 2011; Zhao et al, 2011). The excessive accumulation of F-Cho in hepatocytes leads to endoplasmic reticulum stress, mitochondrial dysfunction, the generation of toxic oxysterols, and the crystallization of cholesterol in lipid droplets and results in the apoptosis, necrosis, and pyroptosis of hepatocytes (Wree et al, 2014; Lebeaupin et al, 2018). The activation of Kupffer cells (liver-resident macrophages) and HSCs by hepatocyte signaling and cholesterol loading also contributes to inflammation and causes liver fibrosis (Beaven et al, 2011).

miR-33a is an intronic microRNA located within the *SREBF2*, a master regulator of cholesterol (Horie et al, 2010a). In contrast, miR-33b is present in large mammals, including humans, within an intron of *SREBF1*, a triglyceride regulator, but is absent in rodents (Horie et al, 2014). We previously generated miR-33b KI mice (a human model) carrying both miR-33a and miR-33b and showed that miR-33a/b, like the host genes, are important regulators of lipid metabolism (Horie et al, 2014). Indeed, compared with WT mice, miR-33b KI mice have lower HDL-C levels and more severe atherosclerosis or aortic aneurysm formation resulting from the reduced expression of the cholesterol transporter ABCA1 (Nishino et al, 2018; Yamasaki et al, 2022). The inhibition of miR-33a/b has been shown to improve HDL biosynthesis in the liver and increase blood HDL-C levels in nonhuman primates (Rayner et al, 2011a). In the present study, miR-33b KI mice exhibited NAFLD/NASH pathology when subjected to a HFD or a GAN diet. Although the NAFLD/NASH in HFD- or GAN diet–fed miR-33b KI mice might have multiple causes, the primary cause is considered to be miR-33b in hepatocytes, as evidenced by the significant increase in hepatic *Srebf1*/miR-33b levels on the GAN diet and our analysis of *Alb*-Cre/miR-33b^fl/fl KI mice. Liver miR-33a was previously found to be important for liver fibrosis (Price et al, 2021); however, it is difficult to evaluate the importance of miR-33b because it was absent in the mice used in this study. The expression of miR-33b was markedly higher than that of miR-33a in the liver of miR-33b KI mice and in human primary hepatocytes from the FANTOM5 database (Andersson et al, 2014; Koyama et al, 2019), and it was considerably induced by the GAN diet, indicating that miR-33b is likely to have a significant effect on pathogenesis. Indeed, the expression levels of

ABCA1 and CPT1A, the target genes of miR-33, are clearly decreased in miR-33b KI mice, causing F-Cho and triglyceride accumulation that leads to hepatic lipotoxicity. Decreased ABCA1 levels can activate multiple inflammatory signals and exacerbate inflammation and fibrosis, as reported previously (Nishiga et al, 2017; Price et al, 2021). Metabolomics analysis in the present study revealed that the levels of palmitic acid, stearic acid, and cholesterol were elevated in the livers of miR-33b KI mice, suggesting that inflammatory responses may be triggered by these factors. A significant increase in liver F-Cho and triglyceride content was observed under GAN diet feeding, suggesting that this diet may have elicited further deterioration of the disease state. The expression of *SREBF1* has been shown to be driven by cholesterol via LXR signaling (Liang et al, 2002) or free fatty acids (Kato et al, 2008); therefore, HFD or GAN diet feeding is likely to have positive feedback effects on *SREBF1*/miR-33b expression. In contrast, macrophage miR-33 has been reported to play a central role in atherosclerosis (Zhang et al, 2022) but does not appear to play a major role in NAFLD/NASH.

We investigated whether antisense oligonucleotides that specifically inhibit miR-33a and miR-33b could be used to treat NAFLD/NASH. The AMOs used in the present study were prepared using amido-bridged nucleic acid, which exhibits improved nuclease resistance and base-specific hybridization compared with conventional locked nucleic acid (Yamamoto et al, 2015) and can discriminate between miR-33a and miR-33b with a difference of only two bases (Yamasaki et al, 2022). In addition, the amido-bridged nucleic acid has been reported to reduce hepatotoxicity, suggesting that the cross-linked nucleic acids are optimal for the treatment of liver diseases (Yamamoto et al, 2015). When these nucleic acid drugs were administered to a mouse model of NASH with miR-33b, they strongly inhibited the accumulation of F-Cho and triglycerides in the liver and improved fibrosis by specifically inhibiting miR-33b. In addition, administering these AMOs caused no specific side effects, suggesting the potential of anti-miR-33b AMO for use in humans.

We also investigated whether anti-miR-33b AMO has therapeutic effects on HSCs. The expression of genes related to fibrosis was suppressed in our in vitro experiment, suggesting that the aforementioned in vivo effects may be partially mediated by effects on HSCs. High cholesterol diet exacerbated liver fibrosis via accumulation of F-Cho in HSCs (Teratani et al, 2012). Accumulation of F-Cho in HSCs was shown to increase the expression of Toll-like receptor 4 (TLR4) and sensitize HSCs to TGF-β signaling, and inhibition of miR-33a ameliorated the F-Cho–induced increase in TLR4 expression (Tomita et al, 2014). Therefore, the anti-fibrotic effects of anti-miR-33b AMO in HSCs may be due to the decrease in F-Cho via increased ABCA1 expression. Because miR-33b is more highly expressed than miR-33a in HSCs and is further induced by TGF-β, anti-miR-33b AMO is expected to have substantial direct beneficial effects on HSCs in vivo.

There are several limitations of this study. Few reports have measured miR-33a and miR-33b in NAFLD/NASH patients (Auguet

and triglyceride levels in the livers of the indicated groups. n = 4–7 samples per group; **$P < 0.01$ and ***$P < 0.001$, one-way ANOVA with Tukey's post hoc test. **(E)** Representative images of cholesterol crystals with and without a polarizing filter in the liver sections of the indicated groups. **(F)** Quantification of cholesterol crystal in the liver sections. n = 4–6 samples per group; **$P < 0.01$ and ***$P < 0.001$, one-way ANOVA with Tukey's post hoc test.

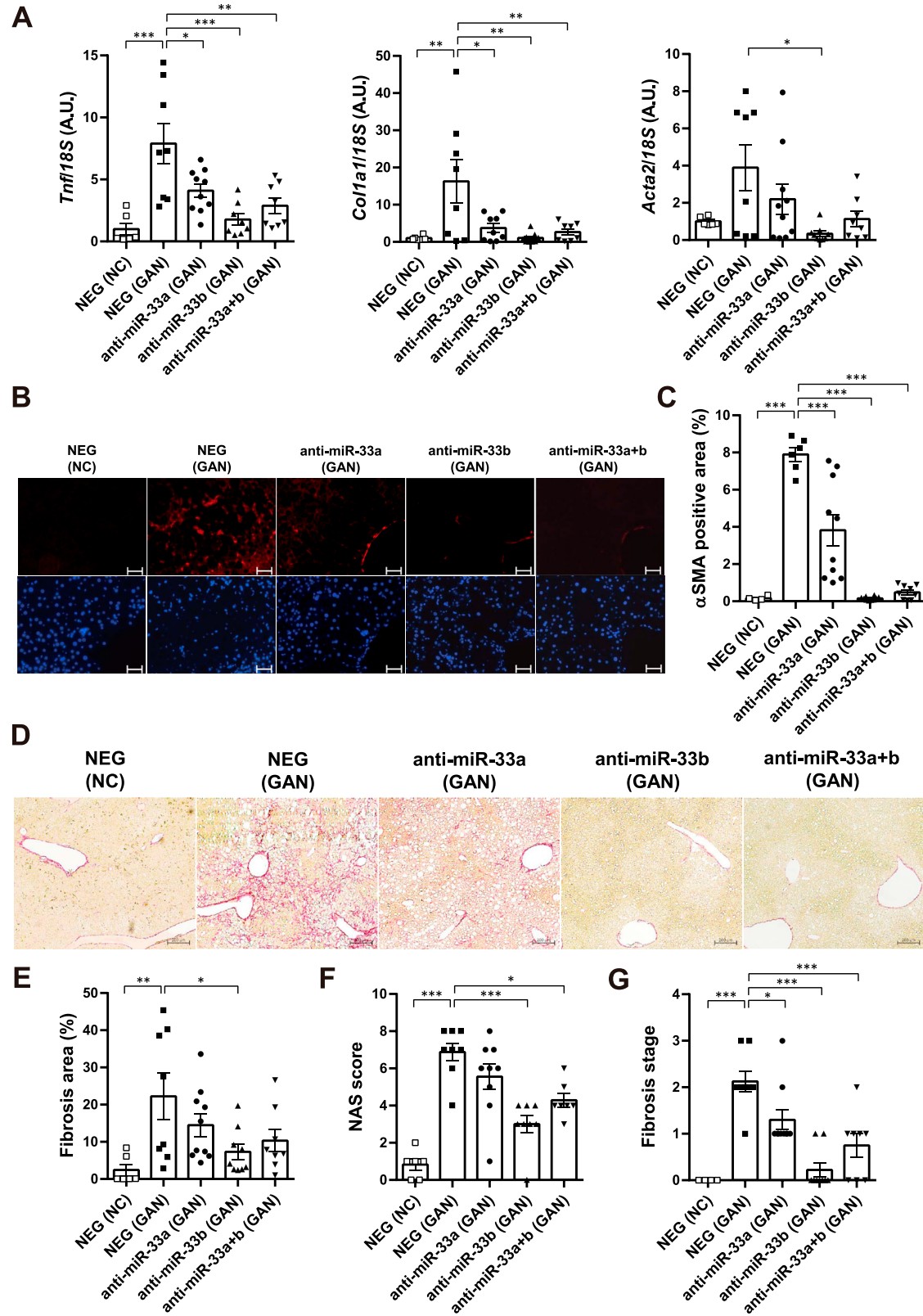

**Figure 5. Anti-miR-33 treatment improves GAN diet–induced inflammation and fibrosis.**
**(A)** Relative expression levels of *Tnf*, *Col1a1*, and *Acta2* in the livers of the indicated groups. n = 6–10 samples per group; *P < 0.05, **P < 0.01, and ***P < 0.001, one-way ANOVA with Tukey's post hoc test. **(B)** Representative microscopic images of immunofluorescence staining for αSMA (red) and DAPI (blue). Scale bars: 100 μm. **(C)** Quantification of the αSMA-positive area in the liver sections of the indicated groups. n = 4–10 samples per group; ***P < 0.001, one-way ANOVA with Tukey's post hoc

et al, 2016; Erhartova et al, 2019). However, there are papers that have examined the expression of *SREBF2* and *SREBF1*, which have miR-33a and miR-33b in their introns, in NASH patients. It has been reported that SREBP2 expression and protein, but not SREBP1c, are increased in human NASH (Caballero et al, 2009; Min et al, 2012). On the one hand, *SREBF1* has been shown to be increased in fatty liver rather than NASH (Dorn et al, 2010; Pettinelli & Videla, 2011). On the other hand, NGS analysis has recently shown that the expression of both *SREBF1* and *SREBF2* is elevated in human NASH samples (Azzu et al, 2021; Yang et al, 2021). Although these are not a comparison between *SREBF1* and *SREBF2* copy numbers and there is no examination of which is more abundant, it is likely that miR-33b and miR-33a are similarly up-regulated in human NASH samples. On the contrary, however, there may be differences in gene regulation between mice and humans. Also, because it has been reported that the *Srebf2* pathway and the resulting cholesterol flux contribute to the generation of LXR agonists required for the transcriptional activity of *SREBP1c* (Rong et al, 2017), differences in diet may create differences between human NASH and mouse models.

In conclusion, using several genetic mouse models, we found that miR-33b in hepatocytes is critical for NAFLD/NASH formation. Targeting miR-33b with AMOs may be an effective treatment for NASH, for which there is currently no established therapy. Inhibition of miR-33b in hepatocytes is clearly important, but future research on the effects of anti-miR-33b AMO on HSCs and the development of a targeted cell-specific drug delivery system may be necessary.

# Materials and Methods

### Animals

C57BL/6J background male mice were used for all in vivo experiments. miR-33b KI mice (miR-33b$^{+/+}$) were generated as reported previously (Horie et al, 2014), and miR-33b$^{fl/fl}$ KI mice were generated as described below. WT littermate mice for each genotype were used as controls. After being weaned at 4 wk of age, mice were fed normal chow (NC) until 8 wk of age, after which they were fed a 45% HFD (D12451; Research Diet), GAN diet (D09100310; Research Diet), or NC for the next 12 wk. All mice were maintained in specific-pathogen-free laboratories at Kyoto University Graduate School of Medicine. Primer sequences for genotyping are listed in Table S5.

### Generation of miR-33b$^{fl/fl}$ KI mice

miR-33b$^{fl/fl}$ KI mice were generated via homologous recombination using embryonic stem (ES) cells. The targeting vector was constructed by modifying a bacterial artificial chromosome using a defective prophage λ-Red recombination system (Copeland et al, 2001). A ~200-bp sequence including the human premature miR-33b sequence was inserted into intron 16 of *Srebf1*. In addition, loxP and

FRT sequences were inserted as shown in Fig S2A. The targeting vector was electroporated into C57BL/6J mouse ES cells (CMTI-2; CHEMICON) using Nucleofector (Lonza) and Mouse ES Cell Nucleofector Kit (VPH-1001; Lonza). Positive clones were selected by incubating cells with 200 μg/ml Geneticin (Thermo Fisher Scientific) for 7 d, and homologous recombination was confirmed via Southern blotting (Fig S2B). Successfully recombined ES cells were injected into blastocysts from ICR mice supplied by Unitech, and chimeric mice were bred with C57BL/6J mice to generate F1 mice. The neomycin resistance cassette was removed from the mouse germ line by crossing with CAG-FLPe transgenic mice, which express FLPe recombinase under the control of the CAG promoter (RBRC01834; Riken). The offspring miR-33b$^{fl/wild}$ KI mice without the CAG-FLPe allele were crossed with each other to generate miR-33b$^{fl/fl}$ KI mice. The genotype of miR-33b$^{fl/fl}$ KI mice was confirmed using tail genome PCR (Fig S2C). Primer sequences for the probes used for Southern blotting and genotyping are listed in Table S5.

### Southern blotting

Southern blotting was performed using DIG-High Prime DNA Labeling and Detection Starter Kit II (Roche) in accordance with the manufacturer's protocol. Genomic DNA samples were purified and digested with EcoRI. The primer sequences used to amplify the probe are shown in Table S5.

### Cell culture and reagents

LI90 cells, a human HSC strain, were obtained from the National Institutes of Biomedical Innovation, Health and Nutrition. These cells exhibit characteristics consistent with those of HSCs, including various connective tissue components (i.e., collagen types I, III, IV, V, and VI), laminin, and fibronectin in addition to vitamin A storage and the biosynthesis of tenascin (Murakami et al, 1995). LI90 cells were seeded on collagen type I–coated dishes (Iwaki Asahi Glass) at a density of $1.0 \times 10^5$ cells/ml with Dulbecco's modified Eagle's medium containing 10% FBS. Peritoneal macrophages were obtained from the peritoneal cavity of mice 4 d after an intraperitoneal injection of 3 ml of 3% thioglycollate. The obtained cells were washed, centrifuged at 200*g* for 5 min, and plated at a density of $8 \times 10^5$ cells/ml with RPMI 1640 medium (Nacalai Tesque) containing 10% FBS. For the transfection of AMOs into LI90 cells, Lipofectamine 2000 Transfection Reagent (Thermo Fisher Scientific) was used according to the manufacturer's instructions. The following antibodies were used: an anti-ABCA1 antibody (NB400-105; Novus Biologicals), anti-GAPDH antibody (#2118S; Cell Signaling Technology), anti-β-actin antibody (AC-15; A5441; Sigma-Aldrich), anti-COL1A1 antibody (ab34710; Abcam), anti-CROT antibody (ab103448; Abcam), anti-CPT1A antibody (ab128568; Abcam), anti-α-smooth muscle actin (SMA) antibody (1A4; C6198; Sigma-Aldrich), anti-F4/80

test. **(D)** Representative microscopic images of Picro-Sirius red staining of the livers of the indicated groups. Scale bars: 200 μm. **(E)** Quantification of the fibrosis area in the liver sections according to Picro-Sirius red staining. n = 6–10 samples per group; *$P < 0.05$ and **$P < 0.01$, one-way ANOVA with Tukey's post hoc test. **(F)** NAS in the liver sections of the indicated groups. n = 6–10 samples per group; *$P < 0.05$ and ***$P < 0.001$, one-way ANOVA with Tukey's post hoc test. **(G)** Fibrosis stage in the liver sections of the indicated groups. n = 4–10 samples per group; *$P < 0.05$ and ***$P < 0.001$, one-way ANOVA with Tukey's post hoc test.

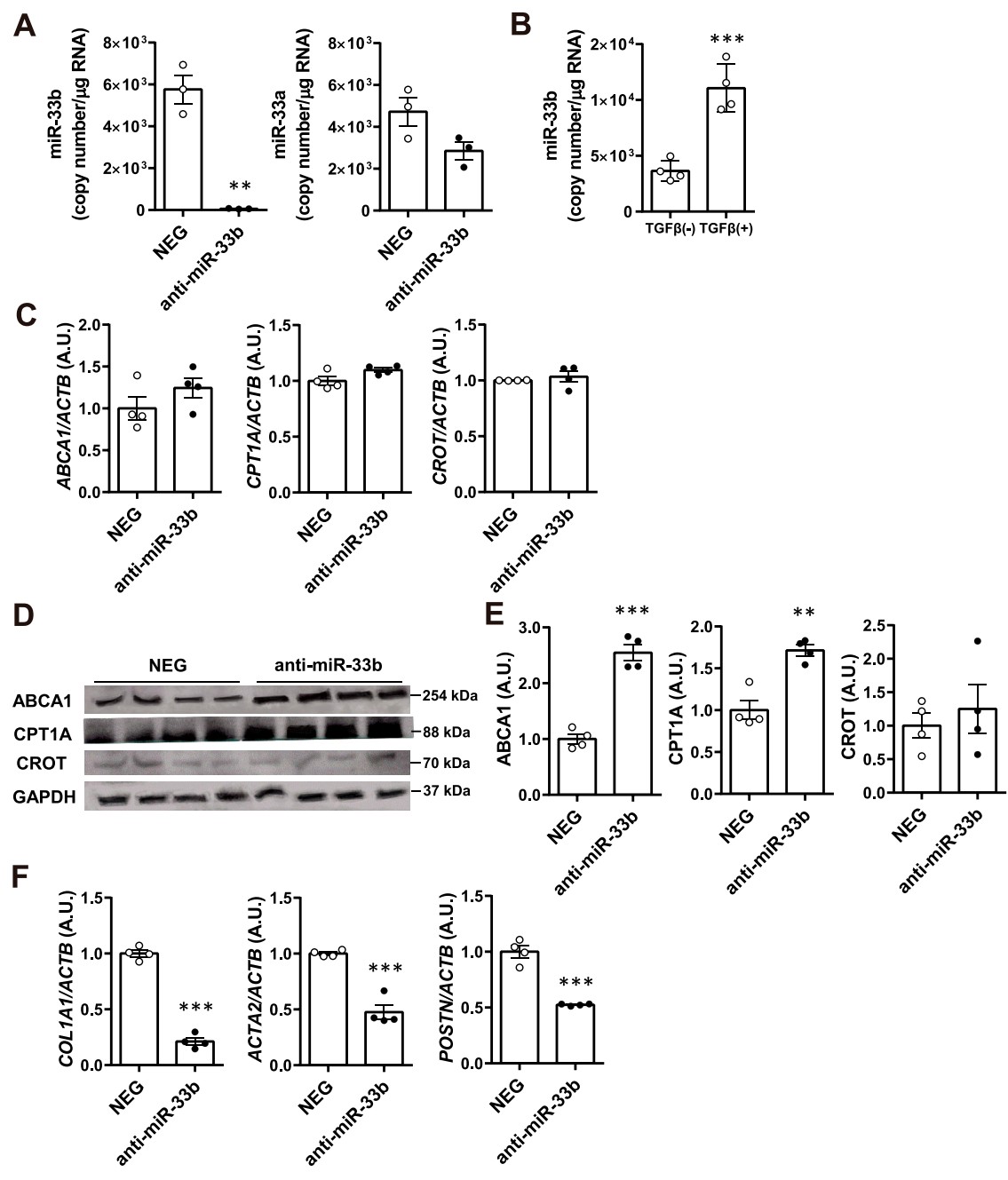

**Figure 6. Anti-fibrotic effect of anti-miR-33b treatment in cultured human HSCs.**
**(A)** Relative expression levels of miR-33b and miR-33a in LI90 cells treated with a negative control or anti-miR-33b. n = 3 cells per group; **$P < 0.01$, unpaired $t$ test.
**(B)** Relative expression levels of miR-33b in LI90 cells treated or not treated with TGF-$\beta$. n = 3 cells per group; ***$P < 0.001$, unpaired $t$ test. **(C)** Relative expression levels of *ABCA1*, *CPT1A*, and *CROT* in LI90 cells treated with a negative control or anti-miR-33b. n = 4 cells per group. **(D)** Western blotting analysis of ABCA1, CPT1A, and CROT expression in LI90 cells treated with a negative control or anti-miR-33b. n = 4 cells per group. **(E)** Densitometry of ABCA1, CPT1A, and CROT in LI90 cells treated with a negative control or anti-miR-33b. n = 4 cells per group; **$P < 0.01$ and ***$P < 0.001$, unpaired $t$ test. **(F)** Relative expression levels of *COL1A1*, *ACTA2*, and *POSTN* in LI90 cells treated with a negative control or anti-miR-33b. n = 4 cells per group.

antibody (BM8; 123101; BioLegend), and anti-myeloperoxidase (MPO) antibody (C-3; sc-390109; Santa Cruz Biotechnology). Anti-rabbit and anti-mouse IgG, HRP-linked antibodies were purchased from GE Healthcare. Anti-rat and anti-mouse Alexa Fluor 488– and 594–conjugated antibodies were purchased from Thermo Fisher Scientific.

## In vivo administration of AMOs against miR-33a and miR-33b

8-wk-old male miR-33b KI mice were used in the following experiments. Negative control (NEG), anti-miR-33a, anti-miR-33b, and anti-miR-33a + anti-miR-33b (anti-miR-33a + b) oligonucleotides were subcutaneously injected into the posterior region of the neck

once every 2 wk (seven injections in total). The nucleic acid doses were at 10 mg/kg; thus, anti-miR-33a and anti-miR-33b were each administered at 5 mg/kg in the anti-miR-33a + b treatment group. 1 wk after the last dose (with mice at 21 wk of age), the animals were euthanized and their blood and organs were collected. A detailed description of the AMOs, including sequences and modifications, was provided previously by Yamasaki et al (2022).

### Serum biochemical analysis

Blood was obtained from the inferior vena cava of anesthetized mice, and serum was separated via centrifugation at 4°C and then stored at −80°C. Using standard methods, biochemical measurements were taken using Hitachi 7180 Autoanalyzer (Nagahama Life Science Laboratory).

### RNA extraction and quantitative real-time PCR analysis of mRNAs

Total RNA was isolated and purified using TriPure Isolation Reagent (Roche), and cDNA was synthesized from 1 μg of total RNA using Verso cDNA Synthesis Kit (Thermo Fisher Scientific) according to the manufacturer's instructions. For quantitative real-time PCR (qRT–PCR), specific genes were amplified in 40 cycles using THUNDERBIRD SYBR qRT-PCR Mix (TOYOBO). Products were analyzed using StepOnePlus (Applied Biosystems), and expression levels were normalized to those of the housekeeping genes *18S* or *Actb*. Gene-specific primers are listed in Table S6.

### qRT–PCR analysis of microRNAs

Total RNA was isolated and purified using TriPure Isolation Reagent (Roche), and miR-33a and miR-33b levels were measured using a TaqMan MicroRNA Assay (Applied Biosystems) protocol. The products were analyzed using StepOnePlus (Applied Biosystems), and expression levels were normalized according to U6 snRNA expression levels. To calculate copy numbers, synthesized miR-33a and miR-33b levels were measured to generate a standard curve.

### Protein extraction and Western blotting

Western blotting was performed using standard procedures described previously (Horie et al, 2010b). In total, 20 μg of protein was fractionated using NuPAGE 4–12% Bis-Tris (Invitrogen) gels and transferred to a PROTRAN nitrocellulose transfer membrane (Whatman). The membrane was blocked using 1× PBS containing 5% nonfat milk for 1 h and incubated with a primary antibody (anti-ABCA1, 1:1,000; anti-CPT1A, 1:1,000; anti-CROT, 1:100; anti-COL1A1, 1:1,000; anti-β-actin, 1:3,000; and anti-GAPDH, 1:3,000) overnight at 4°C. The membrane was washed with PBS containing 0.05% Tween-20 (0.05% T-PBS) and then incubated with a secondary antibody (anti-rabbit IgG, HRP-linked, 1:2,000; anti-mouse IgG, HRP-linked, 1:2,000) for 1 h at 4°C. After the membrane was washed again in 0.05%

T-PBS, ECL Western Blotting Detection Reagent (GE Healthcare) and a LAS-4000 system (Fujifilm) were used for protein detection.

### Measurement of lipid content in the liver

Lipids in the liver were extracted using the Folch procedure (Folch et al, 1957) and quantified using standard enzymatic colorimetric methods.

### Histology and quantification of fibrosis

After mice were administered an overdose of anesthetics, they were perfused with 4% paraformaldehyde. The heart was then excised, and the tissue samples were further fixed with 4% paraformaldehyde at 4°C overnight. The next day, the tissue samples were placed in 70% ethanol for dehydration before they were embedded in paraffin. After the sections were deparaffinized, they were stained with hematoxylin and eosin (HE), Masson's trichrome, and Picro-Sirius red staining. Images were acquired using a microscope (Zeiss). The fibrosis area in the images of Picro-Sirius red staining was measured using image analysis software (ImageJ). The average of five sections from one mouse was taken as the value representing that mouse.

### Immunohistochemistry

Sections of the right liver lobe were stained with anti-αSMA antibody (1:200), anti-F4/80 antibody (1:200), anti-MPO antibody (1:200), and Alexa Fluor 488– and 594–conjugated secondary antibodies (1:200).

### Metabolomics analysis

Mouse liver samples for the indicated genotypes were subjected to capillary electrophoresis–time-of-flight mass spectrometry, capillary electrophoresis–triple quadrupole mass spectrometry, and liquid chromatography–time-of-flight mass spectrometry analyses and metabolite quantification by Human Metabolome Technologies, Inc.

### Statistics

Measurements are presented as means ± SEM. Statistical analyses were conducted using GraphPad Prism 6 (GraphPad Software). Data were compared using paired or unpaired two-tailed $t$ tests and one-way ANOVA with Tukey's post hoc test. $P$-values <0.05 were considered to indicate statistical significance.

### Study approval

This study was approved by the Institutional Review Board of Kyoto University Graduate School and Faculty of Medicine. All animal experimental protocols were approved by the Ethics Committee for Animal Experiments of Kyoto University.

# Life Science Alliance

# Data Availability

The data that support the findings of this study are available from the corresponding authors upon reasonable request.

# Supplementary Information

# Acknowledgements

This work was supported by the Ministry of Education, Culture, Sports, Science and Technology (MEXT) and Japan Society for the Promotion of Science (JSPS) KAKENHI Grants 17K09860 and 20K08904 (to T Horie), and 17H04177, 17H05599, and 20H03675 (to K Ono), as well as a grant from the Cell Science Research Foundation (to T Horie). This research was also supported by AMED under Grant Number 21ym0126013h0001 (to K Ono).

## Author Contributions

S Miyagawa: data curation, formal analysis, and investigation.
T Horie: conceptualization, data curation, formal analysis, supervision, funding acquisition, investigation, and writing—original draft.
T Nishino: data curation and formal analysis.
S Koyama: data curation and formal analysis.
T Watanabe: data curation.
O Baba: investigation.
T Yamasaki: data curation, formal analysis, and investigation.
N Sowa: data curation and formal analysis.
C Otani: data curation and investigation.
K Matsushita: data curation and investigation.
H Kojima: data curation, formal analysis, and investigation.
M Kimura: data curation, formal analysis, and investigation.
Y Nakashima: investigation.
S Obika: data curation and methodology.
Y Kasahara: data curation and methodology.
J Kotera: data curation and methodology.
K Oka: data curation and methodology.
R Fujita: data curation and methodology.
T Sasaki: data curation and methodology.
A Takemiya: data curation and methodology.
K Hasegawa: data curation and investigation.
T Kimura: data curation, supervision, and investigation.
K Ono: conceptualization, formal analysis, supervision, funding acquisition, validation, investigation, project administration, and writing—original draft, review, and editing.

## Conflict of Interest Statement

The authors declare that they have no conflict of interest.

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
