## [Reviewer comments · Life Science Alliance]

Life Science Alliance

Inhibition of microRNA-33b in humanized mice ameliorates nonalcoholic steatohepatitis

Sawa Miyagawa, Takahiro Horie, Tomohiro Nshino, Satoshi Koyama, Toshimitsu Watanabe, Osamu Baba, Tomohiro Yamasaki, Naoya Sowa, Chiharu Otani, Kazuki Matsushita, Hidenori Kojima, Masahiro Kimura, Yasuhiro Nakashima, Satoshi Obika, Yuuya Kasahara, Jun Kotera, Kozo Oka, Ryo Fujita, Takashi Sasaki, Akihiro Takemiya, Koji Hasegawa, Takeshi Kimura, and Koh Ono
DOI: <https://doi.org/10.26508/lsa.202301902>

Corresponding author(s): Koh Ono, Kyoto University

Review Timeline:

Submission Date:	2023-01-05
Editorial Decision:	2023-03-06
Revision Received:	2023-05-02
Editorial Decision:	2023-05-19
Revision Received:	2023-05-23
Accepted:	2023-05-24

Scientific Editor: Novella Guidi

Transaction Report:

March 6, 2023

Re: Life Science Alliance manuscript #LSA-2023-01902-T

Dr. Koh Ono
Kyoto University Graduate School of Medicine
Department of Cardiovascular Medicine
54 Shogoin-Kawahara-cho
Sakyo-ku
Kyoto 606-8507
Japan

Dear Dr. Ono,

Thank you for submitting your manuscript entitled "Inhibition of microRNA-33b ameliorates nonalcoholic steatohepatitis" to Life Science Alliance. The manuscript was assessed by expert reviewers, whose comments are appended to this letter. We invite you to submit a revised manuscript addressing the Reviewer comments.

Thank you for this interesting contribution to Life Science Alliance. We are looking forward to receiving your revised manuscript.

Sincerely,

B. MANUSCRIPT ORGANIZATION AND FORMATTING:

Reviewer #1 (Comments to the Authors (Required)):

In this study, Sawa identified the role of miR-33b in NASH. Due to the absence of miR-33b in mice, the authors generated miR-33b knock-in mice. By using this KI mice, they found that hepatocyte-specific not macrophage-specific miR-33b deletion ameliorated NASH-related fibrosis and inflammation. Importantly, anti-miR-33b treatment limited NASH, suggesting that targeting miR-33b in hepatocytes provides a novel strategy for treatment of NASH. Although this work is interesting, more evidence are needed to support the conclusion.

1. miR-33b is abundantly expressed in human not in rodents. It would be interesting to add evidences from NASH patients for clinical correlation and significance of this study.

2. New targets of miR-33b were not found in this study. The authors should try to find a new target.

3. More experiments are needed to support that anti-miR-33b ameliorates NASH-related inflammation. For example, F4/80 and MPO staining.

Reviewer #2 (Comments to the Authors (Required)):

In their manuscript Miyagawa et al show that genetically engineered mice expressing miR33b within their Srebf1 gene, as naturally occurring in humans, develop steatosis (NAFLD) and features of steatohepatitis (NASH) when fed a high-fat diet (HFD) or Gruba-Amylin NASH diet (GAN). They further attribute this effect to hepatocyte-specific miR33b expression, and not macrophage, through the use of a cell-specific Cre-inducible miR33b knock-in system. Finally, they show that silencing of miR33b but not miR33a via anti-miR oligonucleotide (AMO) administration in miR33b KI mice was sufficient to prevent NAFLD/NASH development in mice showing the specificity of miR33b activity in this phenotype.

The authors have performed a well-rounded study, using what could be described as a top-down approach where they leveraged the power of murine models for in vivo testing of what would be an otherwise primate-specific gene. They successfully generated a Cre-inducible KI of miR33b, used two different models of diet-inducible NAFLD and NASH, and analyzed the development or regression of the phenotype via gold-standard metrics. The use of AmNA-prepared AMO successfully showed specificity between miR33a and miR33b, despite a difference of 2 nucleotides between the two sequences is another impressive feature of the manuscript. Overall, these studies are well performed, the data are clearly presented and the manuscript is well-written.

1. miR33a and miR33b have both been linked to NASH in humans already (Erhartova D. 2019; Auguet T. 2016) and they have the same mRNA targets, as published by the same group (Horie T. 2014). Their effect on NASH seems to rely on the preferential expression of Srebf1 over Srebf2 in this murine model, thereby leading to a higher expression of miR33b. However, the transcriptional activity of SREBPs in humans is known to be distinct from mice and this should be discussed as a limitation of the study.

For instance, in humans, there are reports of SREBP2 but not SREBP1c upregulation in patients with NASH compared to simple steatosis (Caballero F. J Hepatol. 2009), with other reports showing protein but not mRNA upregulation of SREBF2 in NASH, but not NAFLD or obesity (Min HK. Cell Metab. 2012). On the other hand, a moderate increase in SREBF1 has been found in patients with steatosis (below 2-fold), but not NASH (Dorn C. Int J Clin Exp Pathol. 2010). This has been replicated in a cohort of patients with obesity and steatosis but not NASH, with an increase below 2-fold again (Petinelli P. JCEM. 2011). More recently, upstream transcriptional activation analysis by next-generation sequencing in biopsy-proven NASH suggests that both SREBF1 and SREBF2 are active, but with no demonstration of increased transcription of the SREBF1 or SREBF2 locus (Azzu V. Mol Metab. 2021). These findings bring the relevance of the current study into question.

Whether different nutrients in the diets could influence the phenotype should also be discussed. The work of Jay Horton on the contribution of the Srebf2 pathway, and thus of cholesterol flux, to the generation of LXR agonists required for SREBP1c transcriptional activity suggest that differences in cholesterol content in the diet influences the cooperation of both transcription factor, which could also explain differences between human pathology and murine models of diet-induced NAFLD/NASH.

2.

2. Given the concerns raised in point #1, the title should be revised to more accurately represent the work presented by including "humanized mice" - Inhibition of miR-33b in humanized mice ameliorates nonalcoholic steatohepatitis

3. The discussion should also be reorganized to discuss the difference in copy numbers between miR33a and miR33b. Deficiency in miR33a has been implicated in NAFLD/NASH development in mice by the same group (Horie T. 2013), why is targeting of miR33a with AMO not reproducing this phenotype?

4. In the introduction, the authors write that miR-33a inhibition did not reduce atherosclerotic plaque size under hyperlipidemic conditions (marquart et al 2013) but should also cite PMID 26517695 and 23702658 which showed the opposite.

Response to Reviewer #1

We are grateful to Reviewer #1 for the informative and useful comments. As described below, we have considered all of these comments and used them to improve our manuscript.

Reviewer #1:

In this study, Sawa identified the role of miR-33b in NASH. Due to the absence of miR-33b in mice, the authors generated miR-33b knock-in mice. By using this KI mice, they found that hepatocyte-specific not macrophage-specific miR-33b deletion ameliorated NASH-related fibrosis and inflammation. Importantly, anti-miR-33b treatment limited NASH, suggesting that targeting miR-33b in hepatocytes provides a novel strategy for treatment of NASH. Although this work is interesting, more evidence are needed to support the conclusion.

1.miR-33b is abundantly expressed in human not in rodents. It would be interesting to add evidences from NASH patients for clinical correlation and significance of this study.

Thank you very much for your valuable comments. We cited previous papers on human NASH and explained the importance of the present results in NASH patients. We also discussed the possibility that SREBF1 and SREBF2, which are thought to be expressed similarly to miR-33b and miR-33a, may behave differently in mice and humans, and added this point to Discussion.

Inserted sentence (on page 22, paragraph 4, lines 27-page 23, lines 13):

There are several limitations of the study in this paper. Few reports have measured miR-33a and miR-33b in NAFLD/NASH patients (Auguet et al, 2016; Erhartova et al, 2019). However, there are papers that have examined the expression of *SREBF2* and *SREBF1*, which have miR-33a and miR-33b in their introns, in NASH patients. It has been reported that *SREBP2* expression and protein, but not *SREBP1c*, is increased in human NASH (Caballero et al, 2009; Min et al, 2012). While, *SREBF1* has been shown to be increased in fatty liver rather than NASH (Dorn et al, 2010; Pettinelli & Videla, 2011). On the other hand, NGS analysis has recently shown that expression of both *SREBF1* and *SREBF2* is elevated in human NASH samples (Azzu et al, 2021; Yang et al, 2021). Although these are not a comparison of *SREBF1* and *SREBF2* copy number and there is no examination of which is more abundant, it is likely that miR-33b and miR-33a are similarly upregulated in human NASH samples. On the other hand, however, there may be differences in gene regulation between mice and humans. Also, since it has been reported that the *Sreb2* pathway and the resulting cholesterol flux contribute to the generation of LXR agonists required for the transcriptional activity of *SREBP1c* (Rong et al, 2017), differences in diet may create differences between human NASH and mouse models.

2.New targets of miR-33b were not found in this study. The authors should try to find a new target.

Thank you very much for your comments. The most important points of this paper are that mice with the same miR-33b as humans show NASH on a high-fat, high-cholesterol diet and that nucleic acid drugs targeting miR-33a/b are effective against NASH for which there are currently no effective treatments. Although no new miR-33a/b target genes were found in the current paper, we believe that the robustness of the previous papers has been proven.

3.More experiments are needed to support that anti-miR-33b ameliorates NASH-related inflammation. For example, F4/80 and MPO staining.

Thank you very much for your valuable feedback. In this study, F4/80 and MPO staining was performed to confirm the therapeutic effect of anti-miR-33a and anti-miR-33b. The results confirmed that the GAN diet increased F4/80 and MPO-positive cells in the liver, and administration of anti-miR-33a and anti-miR-33b decreased them (Supplementary Figure 7).

We added Supplementary Figure 7 and an explanation in the text on page 12, paragraph 2, lines 5-6 and page 18, paragraph 1, lines 5-8.

Inserted sentence (on page 12, paragraph 2, lines 5-6):

Sections of the right liver lobe were stained with anti- α SMA antibody (1:200), anti-F4/80 antibody (1:200), anti-MPO antibody (1:200) and Alexa Fluor™ 488 and 594-conjugated secondary antibodies (1:200).

Inserted sentence (on page 18, paragraph 1, lines 5-8):

In addition, F4/80 and MPO staining was also performed to determine the effect of anti-miR-33a and anti-miR-33b on inflammation. The results confirmed that the GAN diet increased F4/80 and MPO-positive cells in the liver, while administration of anti-miR-33a and anti-miR-33b decreased them (Supplementary figure 7).

Response to Reviewer #2

We are grateful to Reviewer #2 for the informative and useful comments. As described below, we have considered all of these comments and used them to improve our manuscript.

Reviewer #2:

In their manuscript Miyagawa et al show that genetically engineered mice expressing miR33b within their Srebf1 gene, as naturally occurring in humans, develop steatosis (NAFLD) and features of steatohepatitis (NASH) when fed a high-fat diet (HFD) or Gruba-Amylin NASH diet (GAN). They further attribute this effect to hepatocyte-specific miR33b expression, and not macrophage, through the use of a cell-specific Cre-inducible miR33b knock-in system. Finally, they show that silencing of miR33b but not miR33a via anti-miR oligonucleotide (AMO) administration in miR33b KI mice was sufficient to prevent NAFLD/NASH development in mice showing the specificity of miR33b activity in this phenotype.

The authors have performed a well-rounded study, using what could be described as a top-down approach where they leveraged the power of murine models for in vivo testing of what would be an otherwise primate-specific gene. They successfully generated a Cre-inducible KI of miR33b, used two different models of diet-inducible NAFLD and NASH, and analyzed the development or regression of the phenotype via gold-standard metrics. The use of AmNA-prepared AMO successfully showed specificity between miR33a and miR33b, despite a difference of 2 nucleotides between the two sequences is another impressive feature of the manuscript. Overall, these studies are well performed, the data are clearly presented and the manuscript is well-written.

1. miR33a and miR33b have both been linked to NASH in humans already (Erhartova D. 2019; August T. 2016) and they have the same mRNA targets, as published by the same group (Horie T. 2014). Their effect on NASH seems to rely on the preferential expression of Srebf1 over Srebf2 in this murine model, thereby leading to a higher expression of miR33b. However, the transcriptional activity of SREBPs in humans is known to be distinct from mice and this should be discussed as a limitation of the study.

For instance, in humans, there are reports of SREBP2 but not SREBP1c upregulation in patients with NASH compared to simple steatosis (Caballero F. J Hepatol. 2009), with other reports showing protein but not mRNA upregulation of SREBF2 in NASH, but not NAFLD or obesity (Min HK. Cell Metab. 2012). On the other hand, a moderate increase in SREBF1 has

been found in patients with steatosis (below 2-fold), but not NASH (Dorn C. *Int J Clin Exp Pathol.* 2010). This has been replicated in a cohort of patients with obesity and steatosis but not NASH, with an increase below 2-fold again (Petinelli P. *JCEM.* 2011). More recently, upstream transcriptional activation analysis by next-generation sequencing in biopsy-proven NASH suggests that both SREBF1 and SREBF2 are active, but with no demonstration of increased transcription of the SREBF1 or SREBF2 locus (Azzu V. *Mol Metab.* 2021). These findings bring the relevance of the current study into question.

Whether different nutrients in the diets could influence the phenotype should also be discussed. The work of Jay Horton on the contribution of the *Srebf2* pathway, and thus of cholesterol flux, to the generation of LXR agonists required for SREBP1c transcriptional activity suggest that differences in cholesterol content in the diet influences the cooperation of both transcription factor, which could also explain differences between human pathology and murine models of diet-induced NAFLD/NASH.

Thank you very much for your comment. As you indicated, based on previous papers, there may be differences in the behavior of SREBF1 and SREBF2 between mice and humans, and the following sentence has been added to Discussion as a limitation of this paper.

Inserted sentence (on page 22, paragraph 4, lines 27-page 23, lines 13):

There are several limitations of the study in this paper. Few reports have measured miR-33a and miR-33b in NAFLD/NASH patients (Auguet et al, 2016; Erhartova et al, 2019). However, there are papers that have examined the expression of *SREBF2* and *SREBF1*, which have miR-33a and miR-33b in their introns, in NASH patients. It has been reported that SREBP2 expression and protein, but not SREBP1c, is increased in human NASH (Caballero et al, 2009; Min et al, 2012). While, *SREBF1* has been shown to be increased in fatty liver rather than NASH (Dorn et al, 2010; Pettinelli & Videla, 2011). On the other hand, NGS analysis has recently shown that expression of both *SREBF1* and *SREBF2* is elevated in human NASH samples (Azzu et al, 2021; Yang et al, 2021). Although these are not a comparison of *SREBF1* and *SREBF2* copy number and there is no examination of which is more abundant, it is likely that miR-33b and miR-33a are similarly upregulated in human NASH samples. On the other hand, however, there may be differences in gene regulation between mice and humans. Also, since it has been reported that the *Srebf2* pathway and the resulting cholesterol flux contribute to the generation of LXR agonists required for the transcriptional activity of *SREBP1c* (Rong et al, 2017), differences in diet may create differences between human NASH and mouse models.

2. Given the concerns raised in point #1, the title should be revised to more accurately represent the work presented by including "humanized mice" - Inhibition of miR-33b in humanized mice ameliorates nonalcoholic steatohepatitis

Thank you for your comment. We have changed the title of our manuscript as you suggested.

3. The discussion should also be reorganized to discuss the difference in copy numbers between miR33a and miR33b. Deficiency in miR33a has been implicated in NAFLD/NASH development in mice by the same group (Horie T. 2013), why is targeting of miR33a with AMO not reproducing this phenotype?

Thank you very much for your comments. Recently we have identified the cause of obesity in miR-33a-deficient mice (Nat Commun. 2021 Feb 16;12(1):843.). Our results suggest that loss of miR-33a action in the hypothalamus in miR-33a-deficient mice results in inactivation of sympathetic nerves and brown adipose tissue, leading to obesity in these mice. Since the anti-miR-33a in this paper does not cross the blood-brain barrier, obesity due to miR-33a suppression does not occur.

4. In the introduction, the authors write that miR-33a inhibition did not reduce atherosclerotic plaque size under hyperlipidemic conditions (marquart et al 2013) but should also cite PMID 26517695 and 23702658 which showed the opposite.

Thank you very much for your comments. We have cited the suggested 2 papers and modified the introduction part.

Inserted sentence (on page 6, paragraph 3, lines 17-25):

miR-33a deficiency or inhibition in mice has been shown to increase cellular cholesterol export through upregulation of ABCA1, thereby elevating blood high-density lipoprotein cholesterol (HDL-C) levels and inhibiting atherosclerosis (Horie et al, 2012; Rayner et al, 2011b; Rotllan et al, 2013). Inhibition of miR-33a in mice has also been shown to reduce plaque inflammation and prevent atherosclerosis, in part by promoting M2 macrophage polarization and Treg induction (Ouimet et al, 2015). On the other hand, the inhibition of liver miR-33a increased the blood HDL-C concentration and reversed the cholesterol transport capacity of mice fed a normal diet but did not reduce atherosclerotic plaque size under hyperlipidemic conditions where miR-33a expression was suppressed (Marquart et al, 2013).

May 19, 2023

RE: Life Science Alliance Manuscript #LSA-2023-01902-TR

Dr. Koh Ono
Kyoto University
Department of Cardiovascular Medicine
54 Shogoin-Kawahara-cho
Sakyo-ku
Kyoto 606-8507
Japan

Dear Dr. Ono,

Thank you for submitting your revised manuscript entitled "Inhibition of microRNA-33b in humanized mice ameliorates nonalcoholic steatohepatitis". We would be happy to publish your paper in Life Science Alliance pending final revisions necessary to meet our formatting guidelines.

- please consult our manuscript preparation guidelines <https://www.life-science-alliance.org/manuscript-prep> and make sure your manuscript sections are in the correct order
- please add a conflict of interest statement to your main manuscript text
- please add an Author Contributions section to your main manuscript text
- please use the [10 author names, et al.] format in your references (i.e. limit the author names to the first 10)
- Please upload all figure files as individual ones, including the supplementary figure files; all figure legends should only appear in the main manuscript file
- please remove the highlights section after the Reference section
- please add your main, supplementary figure, and table legends to the main manuscript text after the references section;
- please upload your Tables in editable .doc or excel format;
- Tables should be numbered consecutively with Arabic numerals (1, 2, 3, 4); They can be included at the bottom of the main manuscript file or be sent as separate files.
- please add a callout for Supplementary Figure 7A,B,C,D and Table 7 to your main manuscript text

A. FINAL FILES:

-- Summary blurb (enter in submission system): A short text summarizing in a single sentence the study (max. 200 characters including spaces). This text is used in conjunction with the titles of papers, hence should be informative and complementary to the title. It should describe the context and significance of the findings for a general readership; it should be written in the

present tense and refer to the work in the third person. Author names should not be mentioned.

B. MANUSCRIPT ORGANIZATION AND FORMATTING:

Sincerely,

Reviewer #1 (Comments to the Authors (Required)):

The authors provided more data to support the conclusion. This manuscript should be accepted in this current version.

Reviewer #2 (Comments to the Authors (Required)):

The authors have satisfactorily responded my critiques

May 24, 2023

RE: Life Science Alliance Manuscript #LSA-2023-01902-TRR

Dr. Koh Ono
Kyoto University
Department of Cardiovascular Medicine
54 Shogoin-Kawahara-cho
Sakyo-ku
Kyoto 606-8507
Japan

Dear Dr. Ono,

Thank you for submitting your Research Article entitled "Inhibition of microRNA-33b in humanized mice ameliorates nonalcoholic steatohepatitis". It is a pleasure to let you know that your manuscript is now accepted for publication in Life Science Alliance. Congratulations on this interesting work.

DISTRIBUTION OF MATERIALS:

Again, congratulations on a very nice paper. I hope you found the review process to be constructive and are pleased with how the manuscript was handled editorially. We look forward to future exciting submissions from your lab.

Sincerely,
